# Synergistic Effects of Pr_6_O_11_ and Co_3_O_4_ on Electrical and Microstructure Features of ZnO-BaTiO_3_ Varistor Ceramics

**DOI:** 10.3390/ma14040702

**Published:** 2021-02-03

**Authors:** Muhamad Syaizwadi Shaifudin, Mohd Sabri Mohd Ghazali, Wan Mohamad Ikhmal Wan Mohamad Kamaruzzaman, Wan Rafizah Wan Abdullah, Syara Kassim, Nur Quratul Aini Ismail, Nor Kamilah Sa’at, Mohd Hafiz Mohd Zaid, Maria Fazira Mohd Fekeri, Khamirul Amin Matori

**Affiliations:** 1Advanced Nano Materials (ANoMa) Research Group, Faculty of Science and Marine Environment, Universiti Malaysia Terengganu, Kuala Nerus 21030, Terengganu, Malaysia; syaizwadi@gmail.com (M.S.S.); ikhmal007@gmail.com (W.M.I.W.M.K.); syara.kassim@umt.edu.my (S.K.); mariafazirafekeri@gmail.com (M.F.M.F.); 2Materials Synthesis and Characterization Laboratory (MSCL), Institute of Advanced Technology, Universiti Putra Malaysia, Serdang 43400, Selangor, Malaysia; wanrafizah@umt.edu.my (W.R.W.A.); khamirul@upm.edu.my (K.A.M.); 3Faculty of Ocean Engineering Technology and Informatics, Universiti Malaysia Terengganu, Kuala Nerus 21030, Terengganu, Malaysia; 4Department of Physics, Faculty of Science, Universiti Putra Malaysia, Serdang 43400, Selangor, Malaysia; nurquratulainiismail95@gmail.com (N.Q.A.I.); kamilah@upm.edu.my (N.K.S.)

**Keywords:** ZnO-BaTiO_3_, varistor ceramics, electrical properties, Co_3_O_4_, Pr_6_O_11_

## Abstract

This paper investigated the effects of Pr_6_O_11_ and Co_3_O_4_ on the electrical properties of ZnO-BaTiO_3_ varistor ceramics. The Pr_6_O_11_ doping has a notable influence on the characteristics of the nonlinear coefficient, varistor voltage, and leakage current where the values varied from 2.29 to 2.69, 12.36 to 68.36 V/mm and 599.33 to 548.16 µA/cm^2^, respectively. The nonlinear varistor coefficient of 5.50 to 7.15 and the varistor voltage of 7.38 to 8.10 V/mm was also influenced by the use of Co_3_O_4_ as a dopant. When the amount of Co_3_O_4_ was above 0.5 wt.%, the leakage current increased from 202.41 to 302.71 μA/cm^2^. The varistor ceramics with 1.5 wt.% Pr_6_O_11_ shows good nonlinear electrical performance at higher breakdown voltage and reduced the leakage current of the ceramic materials. Besides, the varistor sample that was doped with 0.5 wt.% Co_3_O_4_ was able to enhance the nonlinear electrical properties at low breakdown voltage with a smaller value of leakage current.

## 1. Introduction

A ZnO varistor is a semiconductor ceramic device possessing nonlinear electrical properties for surge protection applications. Due to these traits, a varistor is capable of sensing and absorbing surge energies quickly, protecting electronic circuits and electric power systems from being damaged. In general, a commercial varistor is mostly prepared by mixing and sintering ZnO powder with several types of varistor-forming oxides, such as Bi_2_O_3_, transition metal oxide (V_2_O_5_), rare earth oxide (Pr_6_O_11_) and alkaline earth oxide (BaO) [1,2,3]. The varistor-former additives form a secondary phase and segregate at the grain boundary, thus imparting nonlinear electrical characteristics to the device [4,5]. The varistor properties originate from its microstructure, where within it contains conducting ZnO grains that are enclosed by a thin insulating oxide layer and some secondary phases which form during the sintering process. The nonlinear property of the ZnO varistor is also known to be dependent on the minor dopants, sintering condition, and final configuration of microstructure [6]. The resultant microstructures become the primary factors for the ZnO varistors to exhibit the nonlinear electrical properties with energy handling capabilities [7]. The other resultant microstructure includes grain boundaries and triple junctions of ZnO grains.

In order to manufacture varistors with good performance, it is essential to study the influence of different additives on the performance of these ceramics. The electrical characteristics of ZnO varistors are directly related to the grain boundary mechanism since both nonlinearity and breakdown voltage is affected by the ZnO grain and grain boundary [8,9]. In the grain boundary, a large number of electronic defects are segregated at the interface layer, forming a potential barrier and enhanced nonlinearity [10]. It is known that dopant additives can control the varistor behavior, particularly when it has a large ionic radius. The dopant ions create a thin intergranular layer and separate the ZnO grains [11,12]. The circumstance then gives rise to the potential barriers which are located at the interface state, where it acts as a trap for the conduction of electrons in the grain boundaries [13]. In varistor applications, the dopant involves adjusting the defect concentration on the ZnO microstructure. The employment of dopant is significant since the varistor performance is sensitive to the presence of dopant additives, even though in a small proportion of their composition [14,15]. Some dopants have their properties in modifying varistor microstructures by either inhibiting the grain growth in the application of high voltage causing it to have a smaller average grain size, or exhibit grain growth and induced a larger average grain size suitable for low voltage application [16].

The present ZnO based varistors are usually doped with several metal oxides that are inserted into their microstructure. Nevertheless, to date, only a few studies have focused on using the perovskite material as an additive. The use of barium titanate (BaTiO_3_) as the varistor forming oxide has attracted much attention due to its unique properties and broad applications in the electronics industry [17]. In a previous study, the ZnO varistor doped with BaTiO_3_ was found to greatly enhance the nonlinear electrical characteristics [18]. The presence of TiO_2_ within the BaTiO_3_ can considerably improve the grain growth of ZnO, making it suitable for the application of low-voltage varistors [19]. In our previous study, the electrical performance of ZnO varistors was investigated using different BaTiO_3_ contents as an additive and found that the optimum nonlinear coefficient (α) value was obtained from the sample made with 12 wt.% content of BaTiO_3_ [20]. The most desirable properties of a varistor are a high value of nonlinear coefficients (α) and a low value of leakage current density (*J_L_*) [21]. Therefore, it is essential to elucidate the effect of Co_3_O_4_ and Pr_6_O_11_ doping into ZnO-BaTiO_3_ based varistor to improve their performance. It is also recognized that transitional metal oxide such as Co and Mn generally improves the nonlinearity coefficient at lower current density region due to an increase in barrier height by electron trapping [22,23]. Moreover, the use of Pr_6_O_11_ dopant can inhibit grain growth that can manage the generation of ZnO grain during the sintering process by promoting more uniform and compact microstructures [24].

The present study further addresses the effect of Pr_6_O_11_ and Co_3_O_4_ doping on ZnO-BaTiO_3_ based varistor ceramics by varying the dopant content in the formulation. The influence of dopants on the microstructure development and electrical properties of ZnO-BaTiO_3_ binary varistors have been analyzed.

## 2. Materials and Methods

### 2.1. Preparation of Powder and Ceramics

The composition of ZnO ceramics (88 − *x*) wt.% with 12 wt.% BaTiO_3_, and *x* wt.% Pr_6_O_11_/Co_3_O_4_ (*x* = 0.5, 1.0, and 1.5 wt.%) were fabricated via the solid-state reaction method. The step began by mixing all ceramic powders using a planetary ball milling for 3 h. Then, the compositions were combined with 1.75 wt.% polyvinyl alcohol (PVA) binder, and the slurry was dried at 80 °C for 2 h. The dried powders were pulverized using an agate mortar and granulated by sieving to produce a starting powder. The resulting powders were next pressed onto ceramic discs or pellets with a diameter of 10.0 mm and about 1.0 mm of thickness at a pressure of 2.6 ton/5 min. The pellets were sintered in a box furnace at 1300 °C for 90 min with a heating and cooling rate of 3 °C/min. Finally, the pellets were polished with sandpapers (P2400 and P4000), and then a silver paste was coated on both sides as an electrode.

### 2.2. Characterizations

The phase content of sample pellets was characterized by X-ray Diffraction (XRD, MiniFlex II, Rigaku, Shibuya, Japan). The surface morphology of the pellets was examined by using a Scanning Electron Microscope (SEM) JEOL JSM-6360LA model (JEOL USA, Peabody, MA, USA), and the elements present on samples were identified using the Energy Dispersive Spectroscopy (EDS). The average density of the sintered samples, *ρ*_avg_ was measured using the Archimedes method, and it was determined based on the following equation:(1)$$\rho_{avg}=\;\frac{m_{air}}{\left(m_{air}-\;m_{water}\right)}\;\times \rho_{water}$$
where $m_{air}$ is the mass of the sintered sample pellet in air, $m_{water}$ is the mass of sample pellet when immersed in water, and $\rho_{water}$ is the density of water used (1.00 g/cm^3^). The average relative density, $\rho_{rel}$ is the ratio of $\rho_{avg}$ to theoretical density, $\rho_{theoretical}$ of sample is evaluated from the expression:(2)$$\rho_{rel}=\frac{\rho_{avg}}{\rho_{theoretical}}\;\times 100\%$$

Electrical characteristics analysis was performed to measure the current density-electrical field (*J–E*) property of varistor samples using a source meter (Keithley 2400, Beaverton, OR, USA). The breakdown voltage (*E_b_*) was determined at a current density of 1.0 mA/cm^2^, while the leakage current density (*J_L_*) was determined at 0.80 *E*_1mA_. The coefficient of nonlinearity (*α*) can be estimated by the following equation:(3)$$\alpha =\;\frac{\log J_{2}\;-\;\log J_{1}}{\log E_{2}\;-\;\log E_{1}}$$
where $J_{1}$ and $J_{2}$ are the current density at the electrical field *E*_1_ and *E*_2_ in the range of 1.0 mA/cm^2^ to 10 mA/cm^2^, respectively [25]. Besides, the nonlinearity can be defined from the relation of $J=KE^{\alpha }$, where *K* is related to the electrical resistivity, and *α* is a nonlinear coefficient [26]. The potential barrier height, $\emptyset_{B}\;$ is an electrical parameter for grain boundaries, which induced the nonlinear properties of varistor ceramics and can be estimated according to,
(4)$$J=AT^{2}\;exp\left[\frac{\beta E^{\frac{1}{2}}\;-\;\emptyset_{B}}{k_{B}T}\right]$$
where *J* is the current density, *E* is the applied electrical field, *A* is the Richardson’s constant (30 A/cm^2^K^2^), *T* is the absolute temperature, β is a constant governed by the relationship *β* ∝ 1/(*rω*), where *r* is grains per unit length and *ω* is the potential barrier width and $k^{B}$ is the Boltzmann constant (8.167 × 10^−5^ eV/K). The value of potential barrier height, $\emptyset_{B}$ can be estimated by plotting a graph of ln *J* versus *E*^1/2^ which was deduced from the intercept of the linear fit of ln *J* vs. $\sqrt{E}$. The lineal intercept method was utilized to obtain the average grain size (*d*) of the microstructure [27].

## 3. Results and Discussions

### 3.1. Scanning Electron Microscopy (SEM)

Figure 1 illustrates the morphology of ZnO-BaTiO_3_ based varistor doped with different contents of Pr_6_O_11_ and Co_3_O_4_. These SEM images show that the surface morphology for varistors with a different dispersion of dopant content induced varied average ZnO grain sizes. For comparison, the microstructure of ZnO-BaTiO_3_ is depicted in Figure 1a. Figure 1a shows that the ceramics have a two-phase microstructure with ZnO grain as the primary phase and BaTiO_3_ phase as the intergranular phase. These phases in the microstructure concurred with the observation by XRD analysis in Figure 2a and Figure 3a at 0.0 wt.% of Pr_6_O_11_ and Co_3_O_4_. The highest peaks match well with the ZnO phase, and the lowest peaks have been matched to the BaTiO_3_-rich phase. A high proportion of BaTiO_3_ rich phase shows that many BaTiO_3_ particles were diffused into some ZnO grains to form an intergranular layer. Another amount segregates near the grain boundaries forms solid solutions. During the sintering process, a liquid phase occurred in the grain boundaries and formed a thick layer along the grain boundary and triple junctions of matrix grains. The remaining amount of solid solutions shows that many BaTiO_3_ particles formed into melt and filled the grain boundaries with a liquid phase during sintering. When the grain boundaries are completely formed at the interface state, no more liquid phase can occur in the grain boundaries. It can be attributed to the precipitation of BaTiO_3_-rich phase near the grain boundaries and nodal points. It is believed that an amount of BaTiO_3_ phase remains at the grain boundaries of two differently matrix which can affect the grain boundaries resistivity. In general, the varistor microstructure is composed of ZnO grains and secondary phase, which usually located at the grain boundaries and junctions of multiple grains [28]. Figure 1b–d show the microstructure of ZnO-BaTiO_3_ varistor ceramics doped with 0.5 to 1.5 wt.% of Pr_6_O_11_. It can be seen that the addition of Pr_6_O_11_ increases the size and number of BaTiO_3_ phase with a slight difference in the shape of the particles in comparison with the sample at 0.0 wt.% Pr_6_O_11_. The difference indicates that the precipitation of Pr_6_O_11_ phase located in the grain boundaries leading to the increase of BaTiO_3_-Pr-rich phase in the form of solid solutions. The presence of BaTiO_3_-Pr-rich phase can enhance the nonlinear electrical properties due to the increase in barrier height at the grain boundaries. This tendency can be observed in Figure 1b–d, where the Pr species have been distributed in the grain boundary and at triple points along with the BaTiO_3_-rich phase. Meanwhile, segregated Pr at the grain boundaries can increase the number of grain boundary that is required for improving the varistor breakdown voltage. By increasing Pr-doping amounts to 1.5, the varistor breakdown voltage of the ceramics gradually increased. This finding are in accordance with the current density-electric field (*J-E*) curves where the varistor breakdown voltage (*E_b_*) of the sample evidently increased when the doping amount is increased up to 1.5. Since Pr^3+^ (1.113 Å) are relatively larger ions than Zn^2+^ (0.74 Å), the chance for Pr ions to dissolve in ZnO matrix grains is extremely small and induced the formation of Pr-rich phase by contributing more solid solutions near the grain boundaries and triple point junctions. The microstructures of the Pr_6_O_11_ doping system did not differ so much, but the grain size increases when doped with 0.5 wt.% of Pr_6_O_11_. The average grain size decreased from 2.0 to 1.6 µm by increasing the Pr_6_O_11_ doping amount up to 1.5 wt.%. It has been well accepted that the ZnO grain growth can be suppressed by excessively doping with Pr_6_O_11_ [29].

The addition of Co_3_O_4_ dopant content enhanced the grain size of ZnO compared to Pr_6_O_11_-doped samples, whereas the average grain size increased from 1.5 to 2.5 µm with increasing cobalt dopant. As shown in Figure 1e–g, the addition of Co_3_O_4_ encourages the growth of ZnO grains and promoted densification of varistors. It was observed that ceramics doped with Co_3_O_4_ contained low of solid solutions with more uniform grains. This effect is due to the solubility of Co ions in ZnO grain. The ionic radii of Co^3+^ (0.65 Å) which is approximately the same as Zn ion and could be induced the solubility of Co ions in ZnO lattice. The incorporation of Co_3_O_4_ decreased the size and number of BaTiO_3_ phase with more resolve BaTiO_3_ particles and diffused into grain boundaries in comparison with that of 0.0 wt.% Co_3_O_4_-doped samples. The Co_3_O_4_ dopant has induced Co ions in the interior of ZnO grains and creates interface states and deep bulk traps that are required for the formation of the potential barrier at the grain boundary and enhance the nonlinearity coefficient of ZnO varistor ceramics [30]. Figure 1e–g indicates that Co species have been diffused into grain boundaries and formed BaTiO_3_-Co-rich phase. By increasing Co-doping amounts up to 1.5, reduced the BaTiO_3_-rich phase due to the segregated of Co ions in the grain boundary and generated the interface traps. A part of Co_3_O_4_ dopant diffused into matrix grains by increasing densification and grain growth of ZnO ceramics during the sintering process. From these findings, the Co_3_O_4_ dopant has increased the barrier height while promoted grain growth and densification of ZnO varistor ceramics.

### 3.2. Energy Dispersive Spectroscopy (EDS)

Additionally, the presence of doping elements was indicated by EDS analysis. Table 1 and Table 2 are the confirmed existing elements in the sample surfaces with different Pr_6_O_11_ and Co_3_O_4_ doping contents. It was observed that the element ratio of Pr increased with the increase of dopant content. However, the element ratio of Ba and Ti was discerned to decrease, which indicates that Pr ions can disturb the BaTiO_3_ phase by segregated near the grain boundaries and triple point junctions. Since the Pr ions are larger than Zn ions, the solubility of Pr ions in ZnO grains is low, and it can attribute to the formation of more intergranular layer by segregating into the grain boundaries. Therefore, the presence of Pr element can reduce the Ba and Ti content at the grain boundaries. Incorporation of Pr ions in the grain boundaries, in this case, caused the formation of more solid solutions near the grain boundaries, which can influence the effective grain boundaries of ZnO varistors.

Nevertheless, the increase of element ratio of Co can increase the electrical charge carriers for better conductivity in ZnO grains. The high solubility of Co ions in ZnO grains makes it easier to segregate in the grain boundary layer by generating the interface state traps, which is essential to enhance the nonlinear properties of ZnO varistor ceramics. As shown in Table 2, better distribution of Ba and Ti elements near the grain boundary led to the increment of Ba and Ti contents. It was indicated that segregation of Co species in the grain boundary promotes the formation of a potential barrier for greater nonlinearity in ZnO varistors.

### 3.3. X-Ray Diffraction (XRD)

The XRD study was used to examine the different phases present in the prepared and sintered samples. Figure 2 and Figure 3 show the X-ray diffraction patterns of the as-prepared ZnO-BaTiO_3_ based varistor doped with different contents of Pr_6_O_11_ and Co_3_O_4_ dopants. The XRD patterns revealed the presence of the ZnO main phase, according to the inorganic crystal structure database (ICSD) No. 067849, with a hexagonal wurtzite structure. The additives were observed due to the presence of the BaTiO_3_ phase within the ZnO ceramics, as indicated by the ICSD No. 028851. XRD spectrum demonstrates that the presence of two major phases which were ZnO phase and BaTiO_3_-rich phase that co-existed with Pr-rich phase. The addition of Pr_6_O_11_ dopant increases the number of intergranular phases and induced solid solutions near the grain boundaries. The diffraction peaks belong to the Pr-rich phase are located at (220) peak, and no secondary phase was observed, suggesting that the Pr ions may be doped into ZnO. Figure 2b shows the magnified region of (220) peak, in which the diffraction angle of Pr-doped ZnO-BaTiO_3_ ceramics with different content shifted to smaller angle side in comparison with that of BaTiO_3_-doped ZnO ceramics. This shifting was due to the difference in ionic radii of Pr^3+^ (1.113 Å) and Zn^2+^ (0.74 Å) which cause to the change of ZnO lattice and the observation are also similar with Pr-doped ZnO ceramic systems reported in [31]. Therefore, the peak shift in different Pr-doped ZnO-BaTiO_3_, in this case, indicates that the doping element with a larger ionic radius than Zn^2+^ is replaced at the substitution sites of the ZnO crystal lattice. Further increase of Pr-doping content up to 1.5, the peak position is shifted towards the larger 2θ side, which suggests that the Pr ion has occupied the interstitial sites of ZnO. In addition, the full width at half maximum (FWHM) of diffraction peaks increase with an increase of Pr doping level, which means that the crystallite size decreases with the increase of Pr content. The mean crystallite size can be calculated using Scherrer’s formula
*D* = 0.9λ/*β*cos*θ*(5)
where λ is the wavelength of X-ray radiation (0.15406 nm), *β* is the full width at half maximum of a distinctive peak, and *θ* is Bragg’s diffraction angle. The micro-strain (ε) can be calculated using formula
Micro-strain (ε) = *β*cos*θ*/4(6)

The calculated full width at half maximum (FWHM) value, the crystallite size (*D*) and micro-strain (ε) of all samples was indicated in Table 3. It can be seen that the crystallite size of ZnO decreased from 57.95 nm to 36.94 nm when Pr concentration increased from 0.5 wt.% to 1.5 wt.%. The data indicate that the presence of Pr^3+^ ions in ZnO reduced the growth of crystal grains. The decrease in crystallite size is due to the compressive strain of crystal lattice. It is clear from Table 3 that the micro-strain is increased from 20.90 × 10^−4^ to 21.20 × 10^−4^. The presence of Pr content in ZnO is able to change the reaction rate as well as the growth mechanism of crystal grains. The decrease in crystal size is well understood by the grain growth suppressing effect of ZnO by Pr-doping [24]. The decrease in the XRD peak position is generally observed when the crystallite size increased and then increased due to the decrease in lattice size. With the addition of Pr_6_O_11_, the Pr^3+^ ions may concentrate at the grain boundaries or on the surface of ZnO in the first segregation. Afterwards, an increase of Pr_6_O_11_ content leads to the formation of BaTiO_3_-Pr-rich phase segregation locating on the surface of the sample and in the grain boundary near the junctions between ZnO grains, which will inhibit ZnO grains.

It was observed that the ceramics doped with Co_3_O_4_ contained two-phase microstructure with ZnO as the major phase and BaTiO_3_ phase as the minor phase. Traces of Co-rich phases were also detected where some minor peaks had been matched to Co_3_O_4_. The formation of intergranular phase was also induced by Co-rich phase at the grain boundaries. However, the addition of Co_3_O_4_ decreased the number of observable intergranular phase and enhanced the grain growth compared to the sample doped with Pr_6_O_11_. In the enlarged view of the (331) peak in Figure 3b, the peak gradually moves to a lower angle with the addition of Co_3_O_4_ dopant, indicating increased in crystallite size as a result of Zn^2+^ (0.74 Å) with the smaller radius being substituted by Co^2+^ (0.75 Å). Table 4 shows the value of full width at half maximum (FWHM), the crystallite size (*D*) and micro-strain (ε) with Co-doping content. With increasing Co content from 0.5 to 1.5 wt.%, the crystallite size is decreased from 36.89 to 35.11 nm. The decrease in peak position value in this case is also due to the increased in crystal size and the effect of tensile strain. Based on this evidence, the micro-strain is increased from 21.90 × 10^−4^ to 24.10 × 10^−4^ and exhibited a larger crystallite size in comparison with that of 0.0 wt.% Co_3_O_4_ addition. Moreover, the value of FWHM decreases with the addition of Co_3_O_4_, which indicates that the crystallite size increases with the addition of Co_3_O_4_ dopant. Then the 2*θ* shifted to larger angle side for 1.5 wt.% Co-doped ZnO-BaTiO_3_ ceramic materials are attributed to the existence of Co_3_O_4_ in ZnO. Therefore, the crystal grains decreased with a further increase of Co-doping content up to 1.5 wt.%.

Moreover, some minor peaks could be observed where their peaks are located in the closed angles of the ZnO and BaTiO_3_ phases. The presence of a new phase might be due to the addition of Pr_6_O_11_ and Co_3_O_4_ dopants, but some phases could not be detected in all samples, which indicates that Co and Pr ions were completely diffused into ZnO and BaTiO_3_ phases. The existence of these phases promotes the formation of intergranular layer and solid solutions at the ZnO grain and grain boundary, which also acts as a donor that forms the depletion layer or as an acceptor by dominating the grain boundary layer [32,33]. Furthermore, the presence of these solid solutions has significant effects on the nonlinear electrical properties of the varistor ceramics.

### 3.4. Current Density-Electric Field (J-E) Characteristics

A preliminary test of electrical properties on finalizing the composition of the ZnO-BaTiO_3_ based varistor was characterized by nonlinearity measurement [20]. The samples added with 12 wt.% BaTiO_3_ showed a relatively higher barrier height than other samples with dissimilar BaTiO_3_ contents; the result is consistent with the standard nonlinear coefficient characteristics. The presence of a potential barrier at the grain boundary caused the nonlinearity characteristics of the varistor [34]. Therefore, the presence of grain boundary contributed to the nonlinearity of the varistor ceramics. Figure 4 shows a significant difference in the *J-E* characteristics of the varistor samples doping with various of Pr_6_O_11_. The conduction mechanism is divided into three regions: pre-breakdown region, breakdown region and upturn region.

In the pre-breakdown region, the linear curves of *J-E* response for all samples indicate that the resistivity of the varistor ceramic samples before breakdown voltage is slightly higher than the nonlinear curves of *J-E* response. The *J-E* curves at different concentrations of Pr_6_O_11_ resemble a nonlinear response with a high resistivity region at the left of the breakdown voltage and a low resistivity region at the right of the breakdown voltage. The values of the nonlinear coefficient are decreased for Pr-doped samples, whereas the *J-E* curves are shifted to the right of breakdown voltage by Pr. However, the most important parameter here is the breakdown voltage. By increasing the Pr-doping amounts to 1.5, the varistor breakdown field of the ceramic evidently increased. The breakdown field of the varistor ceramics is related to the ZnO grain size and the applied electric field at each grain boundary [35,36]. The decrease in the average grain size with further increase in the Pr-doping amount can be clarified by the increase in the numbers of grain boundaries between the electrodes, thereby increasing the breakdown voltage and enhanced the nonlinearity of the varistor ceramics. A high nonlinear coefficient leads to the increased breakdown field of the ceramic samples [37]. The increase in the breakdown voltage with further increase in the Pr-doping content also reduced the leakage current, indicating that it occurs at lower current density due to the increase in barrier height at the grain boundaries. The nonlinear electrical properties of the Pr-added samples is lower than that of the BaTiO_3_-doped ceramic for electric fields with strengths lower than 10 V/mm. It has been reported that most of the Pr_6_O_11_-doped ZnO varistor system exhibited low nonlinear coefficient value at a low-voltage range [38].

It can be observed from Figure 5 that the *J-E* curves at two regions with the pre-breakdown region at the low electric field and breakdown area at the high electric field. The sharper the knee of *J-E* curves between the two regions, the higher the nonlinearity coefficient. However, the varistor samples doping with Co_3_O_4_ exhibited higher breakdown voltage in the breakdown region or nonlinear region and showed higher nonlinearity coefficients in comparison with that of BaTiO_3_-doped samples. The varistor’s breakdown field is directly proportional to the grain boundaries number per unit of thickness where the improvement in the nonlinear behavior of the samples is attributed to the formation of the potential barrier at the ZnO grain boundaries. By increasing Co-doping amounts to 1.0, the varistor breakdown voltage of the ceramic apparently decreased first and then gradually increased when the doping amount is increased up to 1.5. The *J-E* curves with different concentrations of Co_3_O_4_ exhibited similar shapes at electric fields below 10 V/mm, however under the varistor voltage at 1 mA (V_1mA_) the varistor samples doping with 0.5 wt.% showed higher breakdown voltage compared to the ceramics doping with 1.0 and 1.5 wt.%. At V_1mA_, the varistor breakdown voltage obviously decreased when the Co-doping amount is increased up to 1.5, thereby decreasing the values of the nonlinear coefficient of the samples. The reduction in nonlinearity coefficient and breakdown voltage with further increase in the Co-doping content could be attributed to the formation of oxygen vacancies at the grain boundary [39]. It was reported that the oxygen species could increase the density of interface states at the grain boundary and enhanced Schottky barrier height [29].

With an increased amount of doped Co_3_O_4_, more Co ions diffused into the ZnO grain and grain boundary which can improve the characteristic of the varistor samples, including the increment of the breakdown voltage (V_1mA_) to a small extent. However, the excessive Co ions enter into ZnO grains and grain boundaries with the BaTiO_3_-rich phase could destroy the original microstructure due to the accumulation of excessive Co ions which depleted the O ions and promoted the formation of O vacancies in the grain boundary. As a result, the density of interface states decreased and reduced the interface barrier height, thus increase the leakage current density within the varistor samples. The electrical parameters for the current density-electric field (*J-E*) of varistor with different amounts of Pr_6_O_11_ and Co_3_O_4_ are summarized in Table 5 and Table 6. The knee of the *J-E* curve at 0.0 wt.% of Pr_6_O_11_ and Co_3_O_4_ only appeared when a lower electric field was applied as shown in Figure 5, which indicates that the ceramic materials exhibited low breakdown voltage. This suggests that ZnO-BaTiO_3_ varistor ceramics can provide excellent protection performance at low-varistor voltage. Figure 4 shows that the *J-E* curve at 0.0 wt.% is disappeared when a stronger electric field was applied. However, with the addition of Pr_6_O_11_ the knee of the curve is appeared when the electric field was increased, indicating an improvement in breakdown voltage of the materials. The enhancement in the nonlinear electrical performance of Pr-doped ZnO-BaTiO_3_ varistor ceramics is suitable for the application in high-voltage varistor systems. Enhancing protection performance for multilayer ceramic varistors (MLCVs) is required since it shows excellent protection performance with low-voltage and small size [8]. Therefore, they are mostly employed in mobile devices. The improvement in nonlinear electrical properties of Co-doped ZnO-BaTiO_3_ varistor systems at low breakdown voltage can provide excellent protection performance for the application in chip varistors.

The relative density of Pr_6_O_11_ dopants was above 96% of the theoretical density of bulk ZnO (5.61 g/cm^3^), and the average grain size was in the range of 1.6 to 2.0 µm. A significant change in the grain size can produce a dense ceramic body with better varistor properties. The varistor breakdown voltage (*E_b_*) of Pr_6_O_11_ dopants increased from 12.36 to 68.36 V/mm, which indicates that doping with Pr_6_O_11_ decreased grain growth. The increase of varistor voltage is due to the ionic radius of rare earth oxides (Pr_6_O_11_) is larger than that of Zn ions. During the sintering process, the interface layer is formed by the segregation of rare earth oxides into the ZnO grain boundary, resulting in the pinning effect and therefore inhibit the grain growth of ZnO varistors [40].

Meanwhile, the leakage current density decreased with the increase of Pr_6_O_11_ doping amounts. This is due to the increase in the density of varistors, where the use of Pr_6_O_11_ doping had restrained the formation of secondary phases and enhanced the densification of the ceramic structure [38]. However, the increases in the leakage current densities in the Pr_6_O_11_ doping samples compared with the undoped sample may be due to the accumulation of excessive dopants has increased the O vacancies and promote the formation of the depletion layer. It is well known that the O species can increase the density of interface states at the grain boundary, thus enhancing the potential barrier height [39,41]. The addition of Pr_6_O_11_ dopant leads to the inhibition of the potential barrier formation in the grain boundary of the varistor. In contrast, the excessive amount of dopant can cause a variation of the electronic state at the ZnO grain boundary and deteriorate the performance of varistor ceramics. Conclusively, the electrical characteristics of ZnO varistors are greatly enhanced with small amounts of rare earth dopants but critically deteriorated when excessive quantities of rare-earth dopants are used [42]. These varistor formers change the property of ZnO varistors by varying the inter-composition ratio of dopants. Through such changes, it will result in a complication for the multi-layer structure of chip varistors by generating a large quantity of the secondary phases with an insulating spinel phase that played no significant electrical role. This phenomenon will then result in the degradation of varistor properties [43].

For the varistors with Co_3_O_4_ dopants, the α value initially increased when Co_3_O_4_ doping amounts were increased up to 0.5 wt.%. Later, the α decreased when Co_3_O_4_ doping amount was increased up to 1.5 wt.%. The performance of varistors depends on the amount of secondary phase at the grain boundaries since the presence of the BaTiO_3_-rich phase can increase the effective grain boundary region and improve its performance. The increase of dopant content has changed these varistor formers, which resulted in a complication for the grain boundary layer structure by generating a large number of secondary phases. It is known that the addition of transition metal oxides such as Co_3_O_4_ can generally improve the nonlinear coefficient at lower current density region by trapping the electrons and increasing the barrier height [44]. The average relative densities of Co_3_O_4_ dopants exceeded 97% of theoretical density, and the average grain size increased from 1.7 to 2.5 µm as the Co_3_O_4_ content was increased from 0.5 to 1.5 wt.%, respectively. Due to the similar ionic radius of Co and Zn, a solid solution of Zn and Co is formed because of the complete substitution of Co in the Zn structure, which neglects the presence of pores. Moreover, when a sample was strongly doped with Co_3_O_4_, it can increase the defect concentration and the density, which leads to a decrease in the porosity [45].

The varistor breakdown voltage initially increased when Co_3_O_4_ doping content was increased up to 0.5 wt.%. Then, when the doping level was increased up to 1.0 wt.%, it decreased. Also, the *E_b_* value decreased after the Co_3_O_4_ doping amount was increased up to 1.5 wt.%. The decrease of these *E_b_* can be clarified by the increase of the average grain size since both are correlated [46]. The decrease of leakage current density (*J_L_*) at 0.5 wt.% Co_3_O_4_ is due to the electrons which come from the grain are trapped by the defects or dopants at the grain boundary. Therefore, the high density of samples doped with Co_3_O_4_ enhanced the varistor performance by restricting the current flow using high resistance and reduced the leakage current density. By increasing the Co_3_O_4_ doping amount up to 1.5 wt.%, the *J_L_* value was increased due to the presence of multiple defects at the grain boundary. The state also deteriorates the surge absorption capabilities by decreasing the number of active grain boundary region [47]. The nonlinear electrical properties are related to the Schottky barrier height at the grain boundary. Therefore, the increase of α value signifies a higher barrier height and consequently reduced the leakage current density of the varistor.

## 4. Conclusions

In conclusion, Pr-doped ZnO-BaTiO_3_ and Co-doped ZnO-BaTiO_3_ varistor ceramics were fabricated via a solid-state reaction method. The effects of different Pr_6_O_11_ and Co_3_O_4_ dopants compositions on the characteristics of ZnO-BaTiO_3_ based varistor ceramics were investigated. SEM images of Pr-doped ZnO-BaTiO_3_ and Co-doped ZnO-BaTiO_3_ with different contents shows that the microstructure was mostly contained a particle in the form of BaTiO_3_-Pr rich and BaTiO_3_-Co rich phase. The results indicate that the additive was well incorporated into ZnO grain and its interface state by segregating the Pr and Co ions near the grain boundaries and triple point junctions. The EDS analysis confirmed the existence of Ba, Ti, Pr, and Co in the ZnO structure. The formation of ZnO, BaTiO_3_, Pr_6_O_11_ and Co_3_O_4_ phases in the prepared samples were shown by XRD analysis. The diffraction peak position of Pr-doped ZnO-BaTiO_3_ and Co-doped ZnO-BaTiO_3_ shifts towards a smaller-angle side due to the substitution of Pr^3+^ and Co^2+^ into the ZnO lattice. The addition of Pr_6_O_11_ and Co_3_O_4_ significantly influences the nonlinear coefficient. However, for both dopants, the effects were diverse. With the addition of Pr_6_O_11_, the nonlinear coefficient value decreased and caused a reduction of the barrier height due to the variety of defects. The excessive amount has lowered the potential barriers, thereby causing the degradation of the active grain boundary. In contrast, the addition of Co_3_O_4_ enhanced the barrier height by providing charge carriers to the grain boundary. The excessive amount of Co_3_O_4_ dopant could be contributed to the oxygen vacancies at the grain boundary and lowered the potential barriers, thereby causing the degradation of the active grain boundary. The leakage current density and varistor breakdown voltage were lower for Co_3_O_4_ than Pr_6_O_11_. Therefore, for the application of low-voltage varistors, the Co_3_O_4_ dopant is proven to improve the nonlinear electrical properties better than the Pr_6_O_11_. The use of Pr_6_O_11_ and Co_3_O_4_ as additives in ZnO-BaTiO_3_ based varistor ceramics has synergistic effects in improving the microstructure homogeneity and enhances their electrical properties for the application in surge arresters.

## Figures and Tables

**Figure 1 materials-14-00702-f001:**
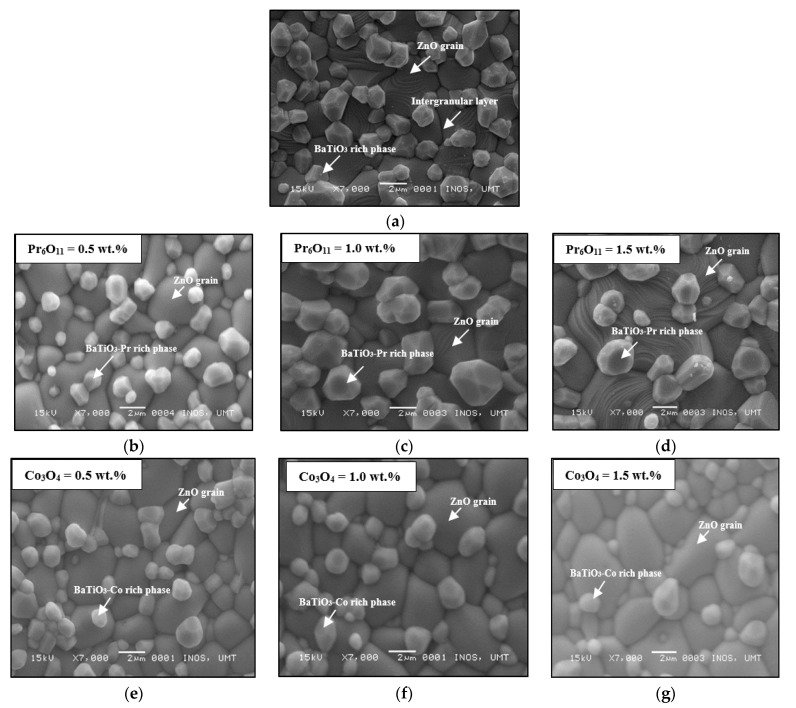
SEM images of ZnO-BaTiO_3_ and doped ZnO-BaTiO_3_ varistor with different concentration of Pr_6_O_11_ and Co_3_O_4_: (**a**) ZnO-BaTiO_3_ without dopant, (**b**) 0.5 wt.% (Pr_6_O_11_), (**c**) 1.0 wt.% (Pr_6_O_11_), (**d**) 1.5 wt.% (Pr_6_O_11_), (**e**) 0.5 wt.% (Co_3_O_4_), (**f**) 1.0 wt.% (Co_3_O_4_) and (**g**) 1.5 wt.% (Co_3_O_4_).

**Figure 2 materials-14-00702-f002:**
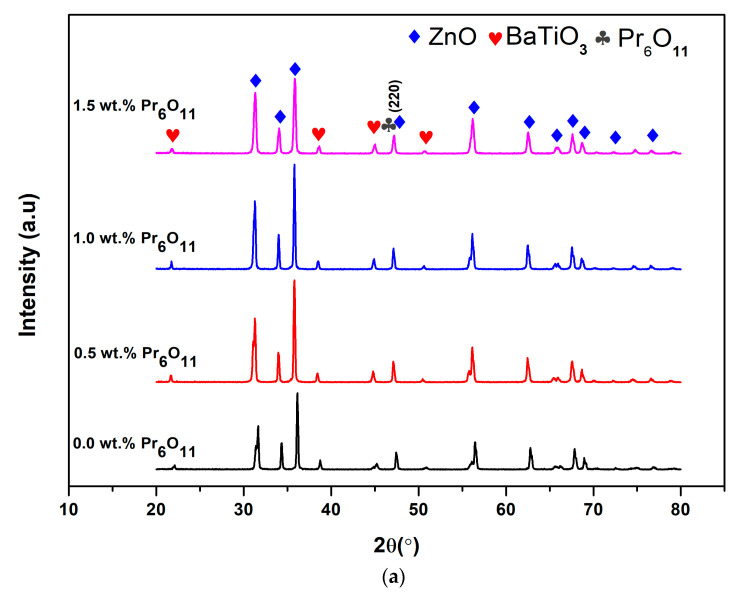
The XRD spectra of ZnO-BaTiO_3_ based varistor doped with different amounts of Pr_6_O_11_ (**a**) XRD patterns of Pr_6_O_11_-doped ZnO-BaTiO_3_ varistor ceramics and (**b**) XRD patterns of (220) diffraction.

**Figure 3 materials-14-00702-f003:**
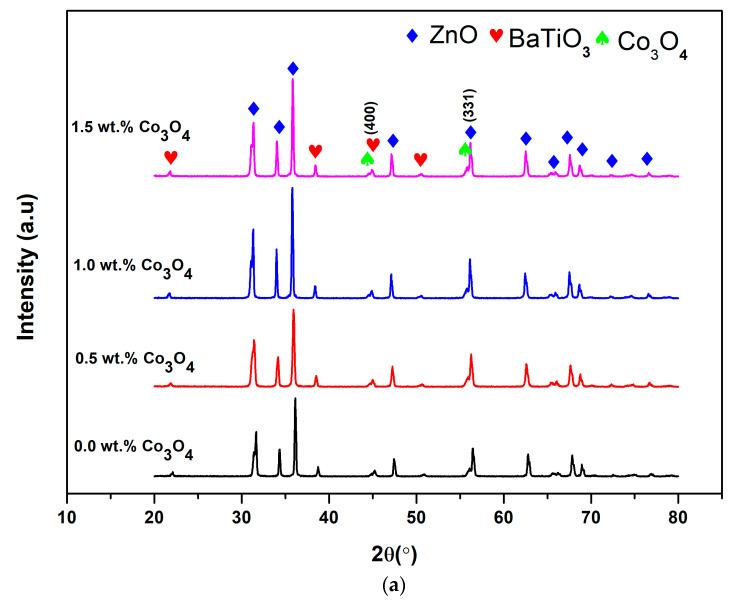
The XRD spectra of ZnO-BaTiO_3_ based varistor doped with different amounts of Co_3_O_4_ (**a**) XRD patterns of Co_3_O_4_-doped ZnO-BaTiO_3_ varistor ceramics and (**b**) XRD patterns of (331) diffraction.

**Figure 4 materials-14-00702-f004:**
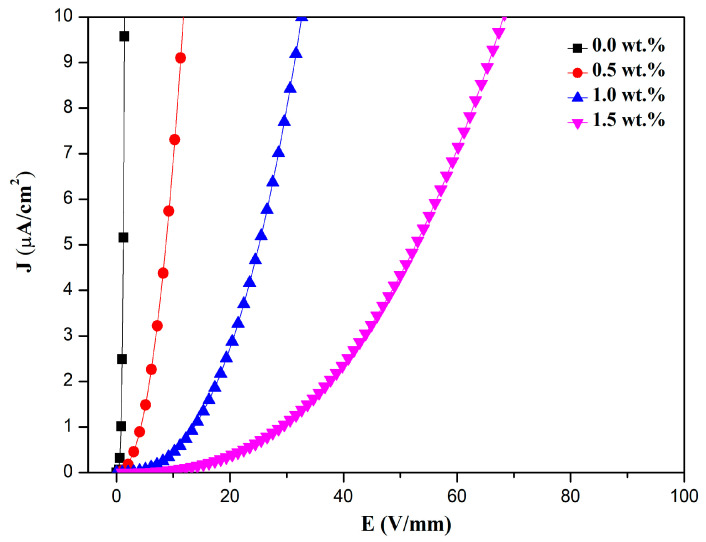
*J-E* characteristics of the varistor ceramics doping with various Pr_6_O_11_.

**Figure 5 materials-14-00702-f005:**
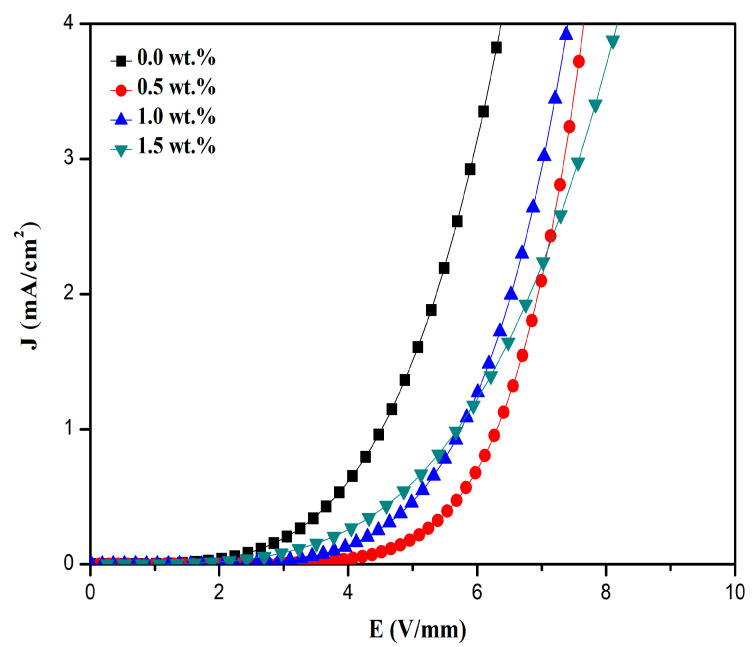
*J-E* characteristics of the varistor ceramics doping with various Co_3_O_4_.

**Table 1 materials-14-00702-t001:** The element contents of ZnO-BaTiO_3_ varistor ceramics with different amounts of Pr_6_O_11_ compositions.

Pr_6_O_11_ Content (wt.%)	Element	Mass (%)
0.5	O K	19.82
	Ti K	6.49
	Zn K	54.65
	Ba L	17.90
	Pr L	1.13
	Total	100.00
1.0	O K	19.82
	Ti K	4.85
	Zn K	61.84
	Ba L	12.20
	Pr L	1.30
	Total	100.00
1.5	O K	19.66
	Ti K	4.28
	Zn K	63.12
	Ba L	11.33
	Pr L	1.62
	Total	100.00

**Table 2 materials-14-00702-t002:** The element contents of ZnO-BaTiO_3_ varistor ceramics with different amounts of Co_3_O_4_ compositions.

Co_3_O_4_ Content (wt.%)	Element	Mass (%)
0.5	O K	20.23
	Ti K	7.52
	Co K	0.37
	Zn K	52.48
	Ba L	19.40
	Total	100.00
1.0	O K	20.42
	Ti K	10.47
	Co K	0.87
	Zn K	40.84
	Ba L	27.41
	Total	100.00
1.5	O K	20.24
	Ti K	8.09
	Co K	0.91
	Zn K	49.51
	Ba L	21.25
	Total	100.00

**Table 3 materials-14-00702-t003:** Variation of FWHM value, crystallite size and micro-strain (ε) of Pr-doped ZnO-BaTiO_3_ varistor ceramics with different content.

Pr_6_O_11_ Content (wt.%)	FWHM, β (Degree)	2θ (Degree)	Crystallite Size (nm)	Micro-Strain, ε (10^−4^)
0.0	0.154	47.43	56.28	21.90
0.5	0.149	47.10	57.95	20.90
1.0	0.168	47.12	51.57	21.20
1.5	0.234	47.18	36.94	9.15

**Table 4 materials-14-00702-t004:** Variation of FWHM value, crystallite size and micro-strain (ε) of Co-doped ZnO-BaTiO_3_ varistor ceramics with different content.

Co_3_O_4_ Content (wt.%)	FWHM, β (Degree)	2θ (Degree)	Crystallite Size (nm)	Micro-Strain, ε (10^−4^)
0.0	0.264	56.48	34.17	21.90
0.5	0.244	56.28	36.89	15.90
1.0	0.245	56.16	36.67	24.10
1.5	0.256	56.20	35.11	20.10

**Table 5 materials-14-00702-t005:** Average grain size (*d*), Relative density, nonlinear coefficient (α), breakdown voltage (*E_b_*), leakage current density (*J_L_*), and Schottky barrier height (Φ*_B_*) of Pr_6_O_11_ at different content.

Pr_6_O_11_ Content (wt.%)	*d* (µm)	Relative Density (%)	*α* ± SE	*E_b_* (V/mm)	*J_L_* (µA/cm^2^)	Φ*_B_* (eV)
0.0	1.5	89.3	4.83 ± 0.16	6.09	348.00	0.88
0.5	2.0	96.8	2.29 ± 0.01	12.36	599.33	0.81
1.0	1.8	97.5	2.66 ± 0.01	33.67	553.33	0.84
1.5	1.6	97.8	2.69 ± 0.02	68.36	548.16	0.86

**Table 6 materials-14-00702-t006:** Average grain size (*d*), Relative density, nonlinear coefficient (α), breakdown voltage (*E_b_*), leakage current density (*J_L_*), and Schottky barrier height (Φ*_B_*) of Co_3_O_4_ at different content.

Co_3_O_4_ Content (wt.%)	*d* (µm)	Relative Density (%)	*α* ± SE	*E_b_* (V/mm)	*J_L_* (µA/cm^2^)	Φ*_B_* (eV)
0.0	1.5	89.3	4.83 ± 0.16	6.09	348.00	0.88
0.5	1.7	99.2	7.15 ± 0.42	7.58	202.41	0.94
1.0	2.2	99.1	5.62 ± 0.18	7.38	294.76	0.92
1.5	2.5	97.9	5.50 ± 0.33	8.10	302.71	0.91

## Data Availability

Data sharing is not applicable to this article.

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
