# Peer review of "Synergistic Effects of Pr6O11 and Co3O4 on Electrical and Microstructure Features of ZnO-BaTiO3 Varistor Ceramics"

_materials, 2021, doi:10.3390/ma14040702_

Round 1
Reviewer 1 Report
After reading the obtained text of proposed paper I have following questions:
- Describing Fig.1a, authors write: „On the surface of the sample, identification of the ZnO grain and BaTiO3 phase were determined, which supports the findings by XRD analysis”. It is not clear how authors obtained such informatiom from XRD data?
- Next we can read that: „Figure 1(a) shows that BaTiO3 phase formed a solid solutions at the grain and grain boundaries of ZnO ceramics”. It is not clear and not consisted with previous phrase.
- In descriptions of figures 1b – 1g we do not read about BaTiO3, but it is used name „secondary phase".
- What means the phrase: „It can be seen that the addition of Pr6O11 increases the solid solutions”?
- I think that in Fig. 2a the fragment between 45-50 degrees should be magnificated , and in Fig. 2b also 55-57 and 65-67 degrees.
- How authors can explain that the data for 0% addition in Fig.3 and Fig.4 are not identical?
As a conclusion I think that the major revision should be done.
Reviewer 2 Report
This paper investigates the synergistic effect of Pr6O11 and Co3O4 on electrical and microstructure features of ZnO-BaTiO3 varistor ceramics. The influence on the characteristics of nonlinear coefficient, varistor voltage, and leakage current is measured by experiments. However, the following points need to be addressed:
- Both Pr6O11 and Co3O4 have effects on the electrical properties of ZnO-BaTiO3 varistor ceramics, the synergistic effect should be illustrated more clearly, the related results should be given under the condition that both Pr6O11 and Co3O4 change together.
- The effects on the electrical properties of ZnO-BaTiO3 varistor ceramics are presented in the manuscript, what about the mechanical and piezoelectric properties? Such as the stiffness and piezoelectric coefficients.
- The higher doping rates of both dopants have significantly determined the microstructure and nonlinear electrical behavior of varistor ceramics. It is to say that the higher the dopant rate is, the better the electrical properties? Is there an optimal dopant rate?
- Doping Pr6O11 and Co3O4 can improve the electrical properties of ZnO-BaTiO3 varistor ceramics, what are the benefits and potential applications? Which should be discussed in the manuscript.
Round 2
Reviewer 1 Report
I think that paper can be accepted for publication